# Contrastive Learning with Adaptive Prompts for Continual Learning

## Abstract

Continual learning (CL) aims to enable models to learn a sequence of new tasks without forgetting previously acquired knowledge. Prompt-based approaches, which adapt small prompt parameters while keeping a large pre-trained backbone frozen, have become a popular strategy to reduce forgetting. However, most existing methods rely solely on visual encoders to effectively guide prompt selection, which leaves them vulnerable to distribution shifts, because biased visual representations can misidentify prompts and lead to severe forgetting. We propose CoLaP, a language-guided prompt selection framework that leverages multimodal models to address this limitation. During training, each input is converted into a rich textual description that provides semantic guidance for training the visual prompt selector. The prompt pool is constructed from clustered concepts that are unique to each dataset, reflecting its specific distribution. In inference, the learned visual selector operates purely on images, preserving efficiency while maintaining the balance between plasticity and stability. Extensive experiments on both in-distribution and out-of-distribution benchmarks show that purely visual prompt methods degrade as the number of tasks grows, whereas our language-informed approach achieves superior generalization and robustness. These results highlight the promise of multimodal semantic guidance for scalable and resilient continual learning.

## 1 Introduction

Continual learning (CL) aims to develop deep models that can acquire novel knowledge across a stream of novel tasks while minimizing *catastrophic forgetting*. Early research in CL primarily addressed this challenge by limiting relevant parameter updates for previous tasks to reduce task forgetting (Kirkpatrick et al., 2017; Aljundi et al., 2018) or by maintaining a replay buffer (Rebuffi et al., 2017; Wu et al., 2019), which allows for the rehearsal of past experiences. However, the rapid emergence of large-scale pre-trained models (Dosovitskiy et al., 2021; Touvron et al., 2023; Radford et al., 2021; Yang et al., 2025) has changed the focus of the research in CL. Current approaches in CL have gone beyond mitigating the distribution shift associated with forgetting and have tackled catastrophic forgetting by exploiting the rich and transferable knowledge embedded in large-scale architectures for continual adaptation.

A common practice for adapting these models is known as prompt-based methods (Wang et al., 2022c;b; Smith et al., 2023), which utilize large pre-trained models by learning a pool of soft-prompts (Lester et al., 2021). These methods learn a set of vectors (prompts) to guide a frozen backbone, and generate concept-adaptive representations that modulate the inference outputs in the presence of novel data inputs. In these methods, the prompt is selected from a pool of alternatives, a critical process to ensure that each prompt specializes for its intended input distribution, ultimately enabling the model to perform task-adaptive inference while keeping the backbone parameters frozen.

The key strength of prompt-based methods is that they keep the backbone parameters frozen, which mitigates forgetting by limiting the parameter updates to the prompts. While this design choice reduces forgetting, it also introduces a critical tradeoff between plasticity and stability (Grossberg, 2013), which hinges on the effective selection of prompts. The prompt selection typically relies on the representations learned on the pre-trained visual encoder; therefore, any biases or misalignments in these features prevent the model from selecting the appropriate prompts. As a consequence, when faced with out-of-distribution data, prompt selection typically under-performs, leading to degraded generalization, interference between tasks, and catastrophic forgetting.

To overcome this limitation, we introduce **C**ontrastive **L**earning with **A**daptive **P**rompts (CoLaP), a language-aligned visual prompt selector for CL. Our key insight is that language models are trained on a vast corpus spanning a broad spectrum of concepts.

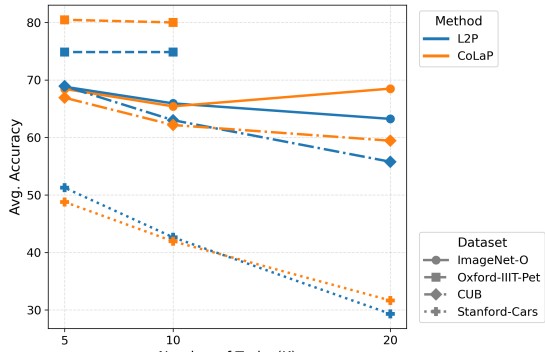

Figure 1: **CoLaP vs. L2P across tasks on four out-of-distribution benchmarks.** Line styles indicate datasets; colors denote methods. CoLaP consistently outperforms L2P as the number of tasks $K$ increases, demonstrating stronger continual learning and robustness against forgetting.

In comparison to visual encoders, language models provide richer and less biased semantic embeddings. In CoLaP we leverage the scale and diversity of language embeddings and devise a prompt selector that learns how to embed visual data in a representation space that aligns with a pre-defined language embedding instead of the original visual embedding. Intuitively, CoLaP serves as a prompting strategy for CL, where the prompt pool is designed to be multi-modal: the keys (which enable the search of suitable prompts) are learned in the language domain, and the values (actual visual prompts) are learned in the visual domain.

When presented with a novel task, CoLaP train over pairs of images and auto-generated captions. The auto-generated text description serves two roles at training time: it guides the selection of visual prompts from the pool (*i.e* the prompt keys in CoLaP are learned in language space), and it supervises a learnable visual prompt selector through contrastive learning. This textual supervision teaches the visual prompt selector how to select adequate prompts keys in the language space despite receiving visual data as input. At inference time, we drop all the language components (text descriptions and language embeddings), and allow CoLaP to operates exclusively over visual data. This design preserves efficiency while maintaining the balance between plasticity and stability, and also avoid any potential leak of semantic information from the language components to the visual predictions. To the best of our knowledge, CoLaP is the first approach to integrate textual representations into the prompt selection stage for CL method that operates over the visual domain.

We evaluate CoLaP on both in-distribution and out-of-distribution CL setups. Figure 1 shows that purely visual prompt-based methods degrade as the number of tasks grows, whereas CoLaP maintains a more stable performance across diverse datasets. These results highlight the benefits of language-informed prompt selection and establish CoLaP as a scalable solution for robust CL.

Our contributions are threefold: **(i) Language-aligned prompt selection.** We introduce CoLaP, the first CL framework that integrates rich textual representations into the prompt selection stage. By supervising a visual prompt selector with contrastive language guidance during training, CoLaP expands the prompt selection space beyond purely visual features. **(ii) Semantic prompt-pool construction.** We propose a novel procedure that generates rich textual descriptions for each input and clusters their embeddings to organize the prompt pool around semantic concepts, allowing synergy between related classes and stronger and more transferable prompts. **(iii) Comprehensive evaluation and state-of-the-art results.** We demonstrate on both in-distribution and out-of-distribution CL benchmarks that CoLaP significantly improves prompt-

selection accuracy, reduces catastrophic forgetting, and enhances robustness to distribution shifts compared with prior prompt-based methods. In particular, CoLaP outperforms L2P and Dual Prompt by an average of 3.6% and 8.7%, respectively, across five out-of-domain datasets where the optimal result for each dataset and method is selected from the 5-, 10-, and 20-task settings (with 5 tasks used for Oxford-IIIT Pet). For forgetting, CoLaP reduces backward forgetting by 2.5% relative to L2P, and achieving comparable forgetting with Dual Prompt. On in-domain tasks (TinyImageNet), CoLaP matches L2P while surpassing Dual Prompt by 4.2% in accuracy, with modest gains in forgetting.

## 2 RELATED WORK

Continual Learning methods have recently been divided into two main groups (Coleman et al., 2025): those that start from random weights and those that use pre-trained models.

**Traditional CL.** State-of-the-art methods in this category can be grouped into Regularisation-Based Methods and Memory-Based Methods. **Regularization-Based methods** aim to preserve parameters that are important for previously learned tasks while allowing the model to learn new ones. The main differences among these approaches lie in how they estimate the relevance of each parameter, with Elastic Weight Consolidation (EWC) (Kirkpatrick et al., 2017) and Memory-Aware Synapses (MAS) (Aljundi et al., 2018) being their primary representatives. While regularization-based methods are effective at mitigating forgetting, they often constrain the model's plasticity, which can reduce its ability to fully adapt to new tasks. **Memory-Based methods**, such as iCaRL (Rebuffi et al., 2017) and BiC (Wu et al., 2019), leverage a buffer memory to store the most representative instances from previous tasks, which are then rehearsed while learning new tasks. These approaches have demonstrated significant improvements over regularization-based methods by better balancing the stability–plasticity trade-off. However, their stability depends entirely on the memory buffer, which can be impractical in some scenarios due to privacy concerns or storage limitations.

**Pretrained-Based Methods.** One of the most popular and closely related to our method are prompt-based approaches. These methods exploit the rich knowledge of large pre-trained models by freezing the backbone and updating only a small set of learnable input instructions (prompts) to adapt quickly to new tasks. Early approaches such as L2P (Wang et al., 2022c) and DualPrompt (Wang et al., 2022b) select prompts according to the visual similarity between an input image representation and a set of learnable keys. Another strategy, exemplified by S-Prompt (Wang et al., 2022a), allocates a dedicated prompt to each task and employs a k-nearest-neighbor to identify the domain at inference. Despite strong results on standard benchmarks, these methods share a fundamental limitation: by relying exclusively on visual representations, they lack the flexibility and generalization needed to handle distributions that deviate from the pre-trained model.

To improve prompt selection, several works explore alternative key mechanisms. CODA-Prompt (Smith et al., 2023) decomposes prompts into learnable components that are dynamically assembled via an attention layer. POP (Hu et al., 2023) and ProgPrompt (Razdaibiedina et al., 2023) progressively concatenate new vectors to the prompt for each task, training only the newly added components while keeping the rest fixed. Other methods complement these ideas with orthogonality regularization at the prompt level to further reduce forgetting (Qiao et al., 2023; Lu et al., 2024).

Closer to our approach, LGCL (Khan et al., 2023) introduces a task-level prompt selection guidance using the textual embedding of the task prompt generated from class names. Although LGCL shows performance improvements, it remains bound to predefined tasks and therefore struggles to share or reuse knowledge across related classes and to generalize to open-ended settings. Our method removes this constraint by employing language guidance in a fundamentally different way. During training, we generate rich textual descriptions for each sample, embed them, and cluster the resulting representations to organize classes semantically. This clustering encourages synergy among similar classes, enabling the model to form stronger, more transferable prompts and to learn relationships that extend beyond rigid task boundaries.

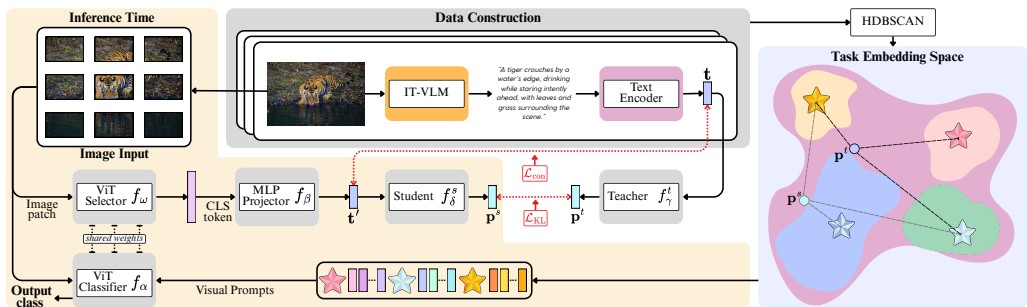

Figure 2: **Overview of the CoLaP framework.** For each task, textual descriptions of training images are embedded and clustered to define the keys of the prompt pool. During training, the student prompt selector uses the frozen visual backbone to predict a distribution over all prompts, aligning visual embeddings with their corresponding textual descriptions via a contrastive loss, and matching the student's predicted distribution to the teacher's distribution (computed from textual embeddings) via a KL-divergence loss. At inference, the top-$K$ prompts are selected for each instance based solely on visual input, concatenated with patch embeddings, and fed through the frozen transformer encoder for classification, enabling task-agnostic and efficient CL.

## 3 LANGUAGE ALIGNED PROMPT POOLS

Contrastive Learning with Adaptive Prompts (CoLaP) is a prompt-based approach for CL that couples visual and language information to improve the prompt generation and selection for pre-trained vision encoders. Figure 2 illustrates the CoLaP framework. CoLaP has two key stages: **(i) Language-Driven Prompt Generation.** For each new task $i$, we embed the natural language descriptions of the training images and generate $Q_i$ clusters from their language embeddings. This clustering step determines the keys of the prompt pool, which are defined as the $Q_i$ cluster centroids in the language space. **(ii) Text-Informed Prompt Selector Training.** CoLaP leverages these cluster assignments to train a set of visual prompts (*i.e* the values of the prompt pool) along with a visual prompt selector whose role is to resolve the approximate language embedding from an input image. This approximate language embedding allows CoLaP to find its top-K most relevant language centroids (the prompt keys) and retrieve the learned visual prompts (the prompt values).

The language-guided prompt pool allows CoLaP to learn and adapt prompts dynamically and align them with the richer semantics encoded in the language embedding, thereby reducing the rigidity of fixed prompt sets and the susceptibility of purely visual methods to distribution shifts, while also reducing catastrophic forgetting.

**Problem definition.** We study class-incremental learning (CIL) for image recognition, here a model $f_\theta$ with parameters $\theta$ is trained sequentially on tasks $\{(X_1, Y_1), \ldots, (X_m, Y_m)\}$. Each task $i$ has image set $X_i$ and disjoint label set $Y_i$ such that $Y_i \cap Y_j = \emptyset$ for $i \neq j$. During inference, the model does not have access to the task identifier. Following standard CIL evaluation(Lopez-Paz & Ranzato, 2017), after learning task $i$ we report the Average Accuracy ($Acc_i$) and Backward Transfer ($BWT_i$):

$$Acc_i = \frac{1}{i} \sum_{j=1}^{i} a_{i,j}, \qquad BWT_i = \frac{1}{i-1} \sum_{j=1}^{i-1} (a_{j,j} - a_{i,j}),$$

where $a_{i,j}$ is accuracy on task $j$ after training up to task $i$. Our goal is to maximize $Acc_i$ while minimizing $BWT_i$, achieving both high performance and minimal catastrophic forgetting.

**Prompt-based Continual Learning.** Formally, let the visual model be the function composition $f_\theta = f_\alpha \circ f_\omega$, where $f_\omega$ is the visual backbone, and $f_\alpha$ is the classifier. The visual backbone $f_\omega$ is a large pretrained Vision Transformer (ViT) that remains frozen throughout continual learning. The backbone can be further decomposed as $f_\omega = f_\phi \circ f_\pi$, where $f_\pi$ is the visual input layer and $f_\phi$ corresponds to the transformer encoder. The input layer $f_\pi$ splits the image $x_i$ into $S$ patches and encodes them to produce $\mathbf{e}_i = f_\pi(x_i) \in \mathbb{R}^{(S+1) \times D}$, where $D$ is the embedding dimension and $S + 1$, accounts the $S$ patch embeddings and the CLS token. To enable rapid adaptation of $f_\omega$ to the incoming data of task $i$, we maintain a global prompt pool $\mathbf{P} = \{P_1, P_2, ..., P_{N_i}\}$, where each prompt $P_j \in \mathbb{R}^{L \times D}$ has length $L$ and $N_i$ denotes the number of prompts until task $i$. For each image $x$, we retrieve the top-K most relevant prompts from $\mathbf{P}$. The selected prompts are concatenated with the patch embeddings before passing through the encoder, $\mathbf{e}^p = [P_{(1:K)}; \mathbf{e}]$, where $P_{(1:K)}$ denotes the ordered set of the top-K prompts and $[.;.]$ indicates sequence concatenation. This augmented representation $\mathbf{e}^p$ is then processed by $f_\phi$, producing an adapted embedding for each instance while preserving the frozen pretrained weights of the backbone and reusing previously learned prompts when relevant.

### 3.1 LANGUAGE-DRIVEN PROMPT GENERATION

For each training task $(X_i, Y_i)$, we first generate detailed textual descriptions $A_i$ for the image set $X_i$ using an Instructional Visual-Language Model (IT-VLM). These language descriptions are obtained with the instruction *"Describe everything in the image, focus on `<CLASS NAME>`"*, where `<CLASS NAME>` is replaced by the name of the class label of $Y_i$. Next, we encode $A_i$ with a state-of-the-art sentence-embedding model to obtain $\mathbf{T}_i$. Let $\mathbf{T}_i = \{\mathbf{t}_z\}_{z=1}^{Z}$ denote the set of language embeddings for the $Z$ images in task $i$. After L2-normalization and dimensionality reduction with UMAP (McInnes et al., 2020), we apply HDB-SCAN (Malzer & Baum, 2020) to obtain $Q_i$ clusters from $\mathbf{T}_i$, where $Q_i \ll Z$. The resulting cluster centroids define a low-dimensional subspace that captures the unique semantic layout of the task $i$. These $Q_i$ centroids constitute the keys of the prompt pool for task $i$. The global prompt pool is updated by simply adding these centroids and their associated set of learnable prompt values. This process yields $N_i = \sum_{j=1}^{i} Q_j$ prompts after task $i$. Importantly, $Q$ is not predetermined. It emerges from the intrinsic structure of the textual embeddings and, therefore, it adapts to the diversity and complexity of each task's language descriptions.

### 3.2 TEXT-INFORMED PROMPT SELECTOR TRAINING

After clustering, we train a cross-modal prompt selector to choose the top-$K$ prompts for each image. The selector is $f_s = f_\delta^s \circ f_\beta \circ f_\omega$, where the frozen backbone $f_\omega$ produces a global visual embedding $\mathbf{v} = f_\omega(x) \in \mathbb{R}^D$, $f_\beta$ projects it to the textual space, $\mathbf{t}' = f_\beta(\mathbf{v}) \in \mathbb{R}^{D_t}$, and the small classifier $f_\delta^s$ outputs a distribution over the current prompt keys $\mathbf{p}^s = f_\delta^s(\mathbf{t}') \in \mathbb{R}^{N_i}$.

**Prompt Selection Space.** Unlike prior prompt-based continual learning methods that maintain continuous key embeddings for each prompt (value) (Wang et al., 2022c;b), CoLaP represents the keys of the prompts only by their categorical identities. Specifically, after task $i$ the global prompt pool contains $N_i$ prompts, and the selector $f_s$ outputs a categorical distribution $\mathbf{k} = \{0, 1, \ldots, N_i - 1\}$ over these indices. Thus each "key" is simply an integer label, not a learnable embedding vector. Therefore, the small network $f_\delta^s$ acts as a classifier, assigning probability mass to each prompt index, from which the top-$K$ prompts are chosen for each input image. This discrete design eliminates the need for storing or updating high-dimensional key embeddings, reducing memory overhead and simplifying continual training.

**Contrastive Alignment Loss.** To align the projected visual representation $\mathbf{t}'$ with the corresponding textual description $\mathbf{t} \in \mathbf{T}_i$, we employ a contrastive loss:

$$\mathcal{L}_{\text{con}} = -\log \frac{\exp\left(\text{sim}(\mathbf{t}', \mathbf{t})/\tau\right)}{\sum_{\mathbf{t}_z \in \mathbf{T}_i} \exp\left(\text{sim}(\mathbf{t}', \mathbf{t}_z)/\tau\right)},$$

where $\text{sim}(\cdot, \cdot)$ denotes cosine similarity and $\tau > 0$ is the temperature.

**Teacher-Student Prompt Prediction.** We introduce a lightweight teacher network $f_\gamma^t$, which receives the text embedding $\mathbf{t}$. This network acts as a text-driven expert in selecting the correct prompts for each instance. So, our small classifier $f_\delta^s$ uses the projected visual vector $\mathbf{t}_i'$ and learns to predict the same prompt distribution as the teacher network.

Both output a distribution over the current global prompt keys:

$$\mathbf{p}^t = \text{softmax}\big(f_\gamma^t(\mathbf{t})\big) \in \mathbb{R}^{N_i}, \qquad \mathbf{p}^s = \text{softmax}\big(f_\delta^s(\mathbf{t}')\big) \in \mathbb{R}^{N_i}$$

The student is trained to match the teacher using a KL-divergence loss:

$$\mathcal{L}_{\text{KL}} = \sum_{n=1}^{N_i} p_n^t \log \frac{p_n^t}{p_n^s}.$$

**Final Objective.** The selector is optimized with $\mathcal{L} = \mathcal{L}_{\text{con}} + \lambda\, \mathcal{L}_{\text{KL}}$, where $\lambda$ balances contrastive alignment and distillation. This training encourages the student to predict language-aligned prompt distributions directly from visual inputs, yielding robust prompt selection even on out-of-distribution data.

## 3.3 INFERENCE PROCESS

At test time, CoLaP relies solely on the trained visual components. Given an input image $x$, the frozen backbone produces a visual embedding: $\mathbf{v} = f_\omega(x)$. The prompt selector, which is composed of the projection head $f_\beta$ and student network $f_\delta^s$, maps $\mathbf{v}$ to a prompt–probability vector over the global prompt pool accumulated during training:

$$\mathbf{p}^s = \text{softmax}\big(f_\delta^s(f_\beta(\mathbf{v}))\big).$$

The top-$K$ prompts from this global pool are concatenated with the patch embeddings and passed through the encoder and classifier for prediction. No textual descriptions, clustering, or teacher network are used during inference, ensuring task-agnostic, fully visual continual learning and minimal computational overhead.

## 4 EXPERIMENTS

To evaluate the robustness of our method, we test it on different benchmarks. As suggested in the motivation, one of the limitations of pre-trained models is the fixed and limited distribution on which these models were trained. For this reason, we evaluate our clustering-based prompt selection framework in both in-domain and out-of-domain benchmarks. Details about datasets and implementation details are in Section A.1 and Section A.2 in the Appendix.

### 4.1 RESULTS AND ANALYSIS

One of the motivations for our proposal is to improve the performance of prompt-based methods by increasing the robustness of the selection mechanism in out-of-domain benchmarks. When we compare CoLaP to approaches that rely on in-distribution and visual-only knowledge, we can observe slightly better performance, as shown in Table 1.

These results suggest that CoLaP is capable of transferring the structure of the textual space into a visual-only space. As a consequence, the visual features are comparable to the in-distribution (language) counterpart.

When comparing CoLaP against purely visual methods in Out-of-domain benchmarks, we can observe a significant difference in performance. We highlight the results in ImageNet-O and Oxford-Pet, where Co-LaP achieves **68.5%** compared to 63.2%, and **80.4%** compared to 74.8%, respectively, representing more than **5%** improvement over L2P. These results can be explained by two key points in our proposal: (1) the low adaptability in the selection process of purely visual methods, and (2) the advantage of using a language embedding during training. Compared to a more recent prompt-based approach, Dual Prompt exhibits substantial instability, as reflected in its high variance (e.g., **42.3% ± 13.4** on ImageNet-O). This indicates that simply extending prompt-based strategies without language guidance does not guarantee robustness. By leveraging language embeddings with rich global knowledge, CoLaP demonstrates both higher accuracy and better stability, making it more suitable for handling domain shifts.

Table 1: **Comparative evaluation of CoLaP against L2P and Dual Prompt baselines across in-domain and out-of-domain datasets.** L2P demonstrates competitive performance on in-domain data, while Dual Prompt exhibits substantial variance and instability across evaluation scenarios. CoLaP consistently achieves superior and stable performance, with notable improvements on out-of-domain tasks attributed to leveraging rich language embeddings. **Bold** indicates best performance; _ denotes second-best results.

| Dataset | L2P | Dual Prompt | CoLaP |
|---|---|---|---|
| **In-domain: TinyImageNet** | | | |
| Acc | 86.17% ± 0.42 | 82.07% ± 0.56 | **86.22% ± 0.79** |
| BWT | 6.40% ± 0.72 | 5.87% ± 0.86 | **5.04% ± 0.81** |
| **Out-of-domain datasets** | | | |
| **ImageNet-O** | | | |
| Acc | 63.24% ± 1.84 | 42.34% ± 13.36 | **68.50% ± 1.82** |
| BWT | 10.81% ± 2.13 | **6.89% ± 2.16** | 9.75% ± 2.49 |
| **Oxford-IIIT-Pet** | | | |
| Acc | 74.88% ± 4.12 | 77.65% ± 3.52 | **80.48% ± 6.22** |
| BWT | 11.92% ± 5.24 | 21.12% ± 3.19 | **10.37% ± 4.91** |
| **CUB** | | | |
| Acc | 55.78% ± 2.26 | 58.74% ± 2.14 | **59.44% ± 1.96** |
| BWT | 19.77% ± 1.84 | **11.14% ± 1.03** | 15.52% ± 2.07 |
| **ImageNet-R** | | | |
| Acc | 67.95% ± 0.76 | 65.01% ± 2.35 | **68.83% ± 0.82** |
| BWT | 7.10% ± 0.91 | **5.13% ± 1.57** | 8.38% ± 0.79 |
| **Stanford-Cars** | | | |
| Acc | 29.33% ± 2.58 | 17.01% ± 6.20 | **31.69% ± 2.71** |
| BWT | 30.90% ± 4.80 | **16.99% ± 2.09** | 23.97% ± 2.12 |

A critical factor underlying these results is the robustness of the prompt selection process. Visual-only methods depend on learning distinct representations to guide selection, but this strategy quickly degrades under out-of-distribution inputs, leading to erroneous choices and increased forgetting. Similarly, prompt-based methods such as Dual Prompt suffer from instability, as their selection remains tied to the visual space without explicit class-aware language guidance. CoLaP addresses both limitations by leveraging language-derived embeddings, which encode broader, less image-biased world knowledge. This general semantic prior improves prompt selection and reduces interference, ultimately enhancing performance and stability across both in-domain and out-of-domain scenarios.

Table 2: **CoLaP performance across continual learning configurations (5-task, 10-task, 20-task) on in-domain and out-of-domain datasets.** Accuracy (Acc) and backward transfer (BWT). Bold indicates optimal performance, ˍ denotes second-best results. CoLaP demonstrates stable accuracy with minimal degradation as task complexity increases, indicating effective retrieval mechanisms.

| Dataset | CoLaP (5 task) | CoLaP (10 task) | CoLaP (20 Task) |
|---|---|---|---|
| **In-domain: TinyImageNet** | | | |
| Acc | 84.98% ± 1.86 | 84.13% ± 1.51 | **86.45 ± 0.50** |
| BWT | 6.61% ± 2.57 | 5.55% ± 1.62 | **5.04 ± 0.81** |
| **Out-of-domain datasets** | | | |
| **ImageNet-O** | | | |
| Acc | 68.45 ± 0.17 | 65.42% ± 1.61 | **68.50 ± 1.82** |
| BWT | **7.44 ± 3.29** | 10.58% ± 2.75 | 9.75 ± 2.49 |
| **Oxford-IIIT-Pet** | | | |
| Acc | **80.48 ± 6.22** | 80.01% ± 5.16 | – |
| BWT | **10.37% ± 4.91** | 10.44% ± 4.90 | – |
| **CUB** | | | |
| Acc | **66.92 ± 3.50** | 62.17% ± 1.72 | 59.44 ± 1.96 |
| BWT | **14.84 ± 3.67** | **14.31% ± 1.77** | 16.11 ± 2.57 |
| **ImageNet-R** | | | |
| Acc | 66.34 ± 2.97 | 66.01% ± 2.21 | **68.16 ± 2.19** |
| BWT | 9.29 ± 4.43 | 8.80% ± 3.11 | **8.28 ± 1.19** |
| **Stanford-Cars** | | | |
| Acc | **48.79 ± 2.62** | 41.94% ± 1.86 | 31.65 ± 2.27 |
| BWT | **10.24 ± 1.78** | 23.16% ± 2.23 | 22.92 ± 3.74 |

## 4.2 ABLATIONS

**Number of tasks.** The number of tasks ins a CL benchmark can significantly impact the performance of the method. Typically, as the number of tasks used increases, the model's performance should decrease, interference and forgetting between specific knowledge become more pronounced. As shown in Table 2, CoLaP exhibits improved robustness as the number of tasks increases, outperforming our baseline methods. We attribute this to the improved quality of representation, specially in data outside of the distribution, we can minimize the interference that occurs when different classes select the same prompt key. With more robust representations, we ensure that the same prompt is used mostly by semantically similar classes, thereby reducing forgetting and improving performance. Table 2 shows that in CUB and ImageNet-O benchmarks, L2P decreases 13.1% and 5.6%, respectively. On the other hand, CoLaP decreases only 7.4% in CUB and maintains its performance in ImageNet-O.

**Number of selected prompts.** An important hyperparameter in our method is the number of prompts retrieved from the global pool. Intuitively, selecting more prompts might capture a wider range of knowledge and improve performance. However, the results in Table 3 show diminishing returns beyond a critical point. A small value such as $K=2$ provides too little guidance for the representation, while larger values (e.g., $K=4$ or $K=5$) introduce redundant or noisy prompts that degrade accuracy and can interfere with other tasks. The best results are obtained at $K=3$, indicating that a careful balance between knowledge diversity and relevance is essential for optimal performance.

Table 3: **Ablation on the number of selected prompts K from the pool.** While increasing K can enrich the knowledge available for representation learning, performance peaks at K=3 and degrades when too many prompts are used, indicating that excessive prompts may introduce noisy or redundant knowledge.

| Dataset | K=2 | K=3 | K=4 | K=5 |
|---------|-----|-----|-----|-----|
| Oxford-IIIT-Pet | 79.31 ± 0.61 | **79.48 ± 3.15** | 78.63 ± 2.20 | 75.73 ± 2.78 |
| CUB | 59.03 ± 0.13 | **59.44 ± 1.96**9 | 58.57 ± 1.26 | 57.60 ± 1.39 |

Table 4: **Ablation on the size of the prompt pool $Q$ under the optimal top-K.** The size of the pool controls the diversity of available prompts: too small a pool limits the representation space, while an excessively large pool makes it difficult to select informative and distinguishable prompts from distant clusters. Performance peaks around $Q = 30$, indicating a balance between diversity and relevance.

| Dataset | $Q$=15 | $Q$=30 | $Q$=50 | $Q$=70 |
|---------|--------|--------|--------|--------|
| Oxford-IIIT-Pet | 77.93 ± 2.74 | **80.47 ± 3.07** | 79.52 ± 2.89 | 78.49 ± 0.39 |
| CUB | 56.04 ± 1.48 | **57.96 ± 1.34** | 56.41 ± 2.99 | 56.48 ± 1.63 |

**Number of clusters.** The number of clusters produced by HDBSCAN determines the prompt–pool size $Q_i$ contributed by task $i$. This value depends on both the distribution of the textual embeddings $T_i$ and the clustering hyperparameters. We tune these hyperparameters to yield approximately $Q \in \{15, 30, 50, 70\}$ clusters per task. As reported in Table 4, pool size strongly influences performance. When the pool is too small (e.g., $Q$=15), prompts must compress diverse semantic information into a limited set of vectors, reducing their capacity to guide representation learning. Conversely, overly large pools (e.g., $Q$=50 or $Q$=70) seem to fragment knowledge across many prompts, making it harder to retrieve informative and increasing the risk of overfitting, as some prompts become overly task-specific. The best results occur near $Q$=30, indicating that a moderately sized pool offers the right balance between diversity and relevance, enabling robust retrieval and generalization.

## 5 CONCLUSIONS

In this paper, we introduce Contrastive Learning with Adaptive Prompts (CoLaP) for Continual Learning. This method aims to enhance the plasticity and robustness of current prompt-based models by employing a multimodal approach. By leveraging the knowledge space of a language model, we expand the corpus and relationships of the representations compared to models trained solely on visual data. CoLaP utilises this knowledge space and proposes a new training strategy to create and select prompts more related to the input regardless of their distribution. This strategy ensures that prompts encompass comprehensive information about the relationships between present and future tasks, facilitating knowledge transfer from the language model to the selection process. As a result, this novel approach to prompt selection makes the model more resilient against out-of-domain benchmarks, improving performance and minimizing forgetting, particularly as the number of tasks increases. Results from multiple benchmarks substantiate this. We believe that using language models as a foundation for training visual models presents a promising research avenue, especially in open-world scenarios.

**Reproducibility Statement** All of the benchmarks that we use are publicly accessible. In Section A.2, we explain all the models and hyperparameters used to replicate our method correctly. To ensure reproducible

results and foster future research, we will made all the resources of this project available upon acceptance. These resources include official weights, training code and benchmark results.

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

# A APPENDIX

## A.1 DATASETS

For the in-domain benchmark, we use one traditionally used in CL: **TinyImageNet** (Han, 2020). This benchmark is a downsampled subset of ImageNet dataset containing 200 object categories with 500 training images and 50 validation images per class at a resolution of 64×64. For the out-of-domain benchmarks, we select models that test the robustness of pre-trained models, either with difficult images from Imagenet or fine-grained datasets. **ImageNet-O** (Hendrycks et al., 2019) is a curated collection of out-of-distribution natural images designed to evaluate robustness and model generalisation beyond the standard ImageNet distribution. **ImageNet-R** (Rendition) (Hendrycks et al., 2021) consists of artistic renditions and stylised variations of ImageNet categories, testing a model's ability to generalise across diverse visual domains. **Oxford-IIIT-Pet** (Parkhi et al., 2012) contains 37 pet breeds with fine-grained class distinctions and large intra-class variability, serving as a benchmark for fine-grained recognition. **CUB-200-2011** (Wah et al., 2011) is a dataset of 200 bird species with rich attribute annotations, commonly used for fine-grained image classification and few-shot learning. **Stanford Cars** (Krause et al., 2013) is a fine-grained classification dataset containing 16,185 images across 196 classes, where each class corresponds to a specific car make, model, and year. The dataset is split into 8,144 training images and 8,041 testing images, exhibiting significant visual similarity between classes, which makes it a challenging benchmark for fine-grained recognition and continual learning.

## A.2 IMPLEMENTATION DETAILS

We employ a frozen large Vision Transformer (ViT) pretrained on ImageNet as our visual backbone and Salesforce-SFR-Embedding-Mistral model (Meng et al., 2024) to obtain the text embeddings $\mathbf{T}_i$. Textual descriptions of images are generated using LLaVA-1.5 (Vicuna7B) Liu et al. (2024). Experimental configurations consider task streams of 5, 10, and 20 tasks. Prompt selection retrieves the top-$K$ prompts with $K = 3$, each of length $L = 5$. The global prompt pool is built from clusters derived from task-specific language embeddings. The number of clusters is not fixed, it emerges from the intrinsic distribution of the embeddings and is influenced by HDBSCAN hyperparameters, which are tuned to yield approximately $Q = 30$ clusters per task. Each task is trained for 5 epochs or until convergence.

In the context of CL, it is common to evaluate different benchmarks using a fixed and reduced number of tasks. However, as we introduce more tasks, the potential for interference between knowledge increases. In this paper, we address this issue by increasing the number of tasks to which the L2P model is exposed, while also highlighting some limitations discussed in our motivation section. We utilize 20 tasks for all benchmarks, except for the Oxford-IIIT-Pet dataset, where we limit the number of tasks to 5 due to the total number of classes. In the ablation study, we demonstrate that our method is more robust when additional tasks are added to the sequence.

Table 5: **Ablation on the depth $F$ of the MLP.** Results on ImageNet-R and TinyImageNet show that performance improves as the depth increases, with the best accuracy obtained at F=3. This suggests that a three-layer MLP is sufficient to capture complex learning patterns, while deeper architectures do not yield additional benefits.

| Dataset | F=1 | F=2 | F=3 | F=4 |
|---|---|---|---|---|
| ImageNet-R | 60.59% ± 0.63 | 61.45% ± 0.27 | **68.16% ± 2.19** | 61.70% ± 2.04 |
| TinyImageNet | 60.88% ± 2.43 | 78.12% ± 0.14 | **86.22% ± 0.79** | 82.15% ± 0.89 |

### A.3 ADDITIONAL EXPERIMENTS

**Depth of MLP** We also ablate on the depth of the MLP layers in CoLaP, as shown in Table 5. Increasing the depth from one to three layers leads to consistent improvements in accuracy, demonstrating that additional capacity helps capture more complex learning patterns. However, extending the depth further to four layers does not yield additional gains, suggesting diminishing returns with over-parameterization. These results indicate that a three-layer MLP provides the right balance between expressiveness and efficiency, making it sufficient to model the semantic structure effectively without unnecessary complexity.

### A.4 THE USE OF LARGE LANGUAGE MODELS (LLMs).

We employed commercial large language models (e.g., ChatGPT, Gemini) only for editorial assistance in refining the manuscript's readability. Their use was restricted to tasks such as grammar correction, clarity improvements, and enhancing the overall flow of the text. They played no part in the study's conception, experimental design, or data analysis.

