# OpenReview forum: "CoLaP: Contrastive Learning with Adaptive Prompts for Continual Learning"
_ICLR.cc/2026/Conference — Submitted to ICLR 2026_

### Official Review · Reviewer_gx7J · 2025-10-22

**Soundness:** 1
**Presentation:** 1
**Contribution:** 2
**Rating:** 2
**Confidence:** 4

**Summary:**

This paper proposes CoLaP, a multimodal framework for continual learning that introduces language-guided prompt selection.  By using textual descriptions from an instruction-tuned vision-language model to form semantically clustered prompt pools and to train a language-aligned visual selector via contrastive learning and distillation. During inference, the model operates purely in the visual domain. Experiments on multiple in-domain and OOD benchmarks super L2P and DualPrompt.

**Strengths:**

1. The paper addresses the problem of out-of-distribution generalization in continual learning from a novel and interesting perspective.
2. The method of leveraging language to guide a visual prompt selector is intuitive and presents a promising research direction for improving robustness.

**Weaknesses:**

1. The method is only compared against two baselines from 2022 (L2P, DualPrompt), while omitting more recent and relevant prompt-based methods mentioned in the related work such as CODA-Prompt, ProgPrompt,  (which are also mentioned in the related works) and DIKI [1]. Comparisons against other families of CL methods, such as regularization-based or LoRA-based approaches such as SD-LoRA[2], are also missing.
2.  While hyperparameters are ablated, there is no ablation analysis of the impact of the KL-divergence loss or the teacher-student distillation framework.
3. The term "Adaptive" in the title is not well-justified or explained in the paper. The abstract and methodology instead focus on "language-guided prompt selection," creating an inconsistency in the paper's framing.
4. The proposed pipeline introduces computational overhead and external dependencies. It requires a large VLM to generate descriptions, another model for text embeddings, and a clustering step. This complexity and the unanalyzed robustness of the generated descriptions may limit the method's practical application.
5. The lack of source code and the generated textual description files raises concerns about the work's reproducibility.

[1] Tang L, Tian Z, Li K, et al. Mind the interference: Retaining pre-trained knowledge in parameter efficient continual learning of vision-language models[C]//European conference on computer vision.

[2] Wu Y, Piao H, Huang L K, et al. SD-LoRA: Scalable Decoupled Low-Rank Adaptation for Class Incremental Learning[C]//The Thirteenth International Conference on Learning Representations.

**Questions:**

1. How do the inference-time FLOPS or latency of CoLaP compare to the baseline methods?
2. Is there a risk of semantic leakage if the generated textual descriptions are overly correlated with the class names, and how is this possibility addressed?

---

### Official Review · Reviewer_jX4A · 2025-10-23

**Soundness:** 2
**Presentation:** 2
**Contribution:** 2
**Rating:** 2
**Confidence:** 4

**Summary:**

This paper presents CoLaP, a language-guided prompt selection framework for continual learning that enhances robustness against distribution shifts. Unlike prior methods that rely solely on visual encoders, CoLaP incorporates textual descriptions during training to provide semantic guidance for prompt selection, enabling more reliable and adaptive representations. A concept-clustered prompt pool captures dataset-specific distributions, while inference remains purely visual to ensure efficiency. By leveraging the knowledge space of language models, CoLaP facilitates better alignment between current and future tasks, improving knowledge transfer and reducing forgetting. Extensive experiments on in-distribution and out-of-distribution benchmarks demonstrate that CoLaP significantly outperforms several purely visual prompt methods in both generalization and scalability.

**Strengths:**

1. The paper introduces a language-guided prompt selection framework that integrates multimodal semantic guidance to address distribution shifts, improving robustness and generalization in continual learning.

2. It provides extensive experimental validation on both in-distribution and out-of-distribution benchmarks to support its claims of improved performance and scalability.

**Weaknesses:**

1. The sentence “CoLaP is the first approach to integrate textual representations into the prompt selection stage for CL method that operates over the visual domain.” appears to overstate the novelty of the contribution, as prior work such as LGCL (Khan et al., 2023) also leverages textual embeddings for prompt selection. Although the authors already discuss LGCL in the related work, this claim in the contribution section should be rephrased to more accurately reflect the distinction between CoLaP and existing methods.

2. The review of traditional continual learning methods is not comprehensive. In addition to regularization- and memory-based methods, architecture-based approaches [a-f] should also be discussed to better contextualize the contribution.
[a] Lifelong learning with dynamically expandable networks, ICLR18
[b] Learn to grow: A continual structure learning framework for overcoming catastrophic forgetting. ICML19
[c] Beef: Bi-compatible class-incremental learning via energy-based expansion and fusion. ICLR23
[d] Overcoming catastrophic forgetting with hard attention to the task. ICML18
[e] Compacting, picking and growing for unforgetting continual learning. NeurIPS19
[f] Parameter-level soft-masking for continual learning. ICML23

3. The notions in Figure 2 should be clarified with a legend or more detailed caption. And the prompt pool appears missing and is suggested to be explicitly annotated in the figure.

4. The definition of “key” is confusing. The authors state that keys are integer labels but also mention clustering centroids as keys. If the intent is that the prompt index is an integer while the key embedding is a fixed (non-learnable) vector, this should be clearly and consistently described.

5. The current method section only introduces the loss for the selector, but does not describe the overall training objective, including how prompts are trained. A more complete description would improve clarity and reproducibility.

6. The statement “The prompt selector, which is composed of the projection head f_\beta and student network f_\delta^s, …” is in consistent with the earlier definition f_s= f_\delta^s \odot f_\beta\odot f_\omega. This discrepancy should be resolved for consistency.

7. The experimental comparison is limited to L2P and DualPrompt. It should also include recent prompt-based methods such as HiDe-Prompt, S-Prompt, CODA-Prompt, ProgPrompt, LGCL, VQ-Prompt, Cprompt, etc. Moreover, the BWT metric underperforms in most cases, which weakens the claim of reduced forgetting and warrants further analysis or discussion.

8. The number of cluster centroids Q is fixed across datasets and tasks, which may limit adaptability. An adaptive strategy could potentially improve performance.

9. Some minors.
1) The font size in Figure 2 is too small and could be increased for readability.
2) Equations are suggested to be numbered for easier reference.

**Questions:**

1. Could the authors rephrase the statement of being the first approach to integrate textual representations into the prompt selection stage to more accurately reflect the nature of their contribution?

2. Could the authors include a more comprehensive discussion of architecture-based methods and other relevant prompt-based approaches to better position CoLaP in the broader CL landscape?

3. Could the authors provide a more complete methodological description that include all loss terms and their interactions to enhance clarity and reproducibility?

4. Could the authors further clarify the keys, specifically whether they are integer indices, fixed embeddings, or learnable representations?

5. Could they reconcile the notation of the selector components for internal consistency for internal consistency across the text?

6. Could the authors report or discuss the performance of CoLaP compared with more recent prompt-based methods such as HiDe-Prompt, S-Prompt, CODA-Prompt, ProgPrompt, and LGCL, which are not currently included as baselines?

7. Since the reported BWT metric underperforms in several cases, could the authors provide additional analysis or justification to support their claim of reduced forgetting, or explain why this metric may not fully capture the benefits of CoLaP?

8. Did the authors explore or consider adaptive or dataset-dependent strategies for determining the number of cluster centroids Q, and if not, could they discuss why a fixed value was chosen?

---

### Official Review · Reviewer_DSrc · 2025-11-01

**Soundness:** 2
**Presentation:** 1
**Contribution:** 2
**Rating:** 2
**Confidence:** 4

**Summary:**

CoLaP argues that multimodal LLMs, having been trained on a large number of concepts, are better suited to encode task information compared to visual-only models that rely solely on visual features for prompt selection. The limited knowledge of visual models can significantly harm accuracy, especially on out-of-domain datasets. CoLaP introduces a novel contrastive learning based method that allows the use of language models by feeding images into a vision–language model to generate text embeddings, then finding cluster centers for these embeddings, which serve as the keys for each task. A classifier generates a categorical distribution over these keys to select the appropriate prompt. The framework further includes a teacher–student network, where the student distribution is trained to match the teachers'. At inference time, the student network selects the top-K prompts.

**Strengths:**

- CoLaP introduces a novel perspective by using multimodal LLM embeddings to train a prompt selector. Unlike ViTs, which are trained on a much smaller set of visual concepts, multimodal LLMs are exposed to a vast range of concepts, reducing semantic misalignments.

- CoLaP proposes a novel framework that leverages multimodal LLM embeddings for prompt selection in downstream tasks.

- CoLaP outperforms vision-based prompt selection methods such as L2P and DualPrompt, particularly on out-of-distribution tasks.

**Weaknesses:**

- Comparison is incomplete. Many later works that can potentially outperform are not compared.
1. RanPAC: Random Projections and Pre-trained Models for Continual Learning
2. Dynamic Integration of Task-Specific Adapters for Class Incremental Learning
3. Adapter Merging with Centroid Prototype Mapping for Scalable Class-Incremental Learning
and many more.
Even within prompt-based methods, authors have not compared with later works such as CODA-prompt (CODA-Prompt: COntinual Decomposed Attention-based Prompting for Rehearsal-Free Continual Learning).

- It is also unclear how exactly the prompts are trained. The paper talks about 'The global prompt pool is updated by simply adding these centroids and their associated set of learnable prompt values,' but does not mention how these 'learnable prompt values are really learnt? Do we train them separately after training the prompt selector? Do we train them in parallel? Do you generate t' using the learnable prompts? Do you simply select the correct task prompts during training and train them using cross-entropy loss? It is unclear

- The main figure is also vague. A key representing meaning of elements (arrows, colors) in the figure would be a nice addition.

- What is the teacher network? Is this an MLP? Is this also trained?

- Table 2 only compares COLLAP results and does not include other methods.

- Reporting average performance would be nice.

- No analysis of generated captions is shown. They could contain errors as well.

- Per task, backward transfer curves and accuracy curves would be nice to look at.

- What is in Table 5? Is this the projector that generates t'

- The paper needs a lot more polishing. Important details are missing. The figure needs to be improved, and more comparisons are needed as well.

**Questions:**

-How are you training the prompt values? Do you also train the old prompts when a new task arrives since the prompt selector outputs distribution over the whole pool at task i? Or do you keep old prompts frozen when a new task arrives? Please be detailed.

- What is in Table 5?

- Why aren't you comparing with the latest methods? How does your method compare with the latest methods?

- Some detailed ablations on design choices of networks used (MLPs,LLaVA and text embedder) will be helpful. Add some analysis of captions and their effect on performance.

- Accuracy curves are also required to show the average accuracy per task during training.

---

### Official Review · Reviewer_EX8F · 2025-11-01

**Soundness:** 2
**Presentation:** 3
**Contribution:** 2
**Rating:** 2
**Confidence:** 5

**Summary:**

The paper introduces CoLaP (Contrastive Learning with Adaptive Prompts), a multimodal continual learning framework that integrates language-guided prompt selection to mitigate catastrophic forgetting. Unlike prior visual-only prompt methods, CoLaP leverages auto-generated textual descriptions to cluster semantically aligned prompts, training a visual selector via contrastive and distillation losses. At inference, it operates purely on visual data, maintaining efficiency. Extensive experiments on in-domain and out-of-domain benchmarks (e.g., TinyImageNet, ImageNet-O) show CoLaP achieves superior generalization and stability, outperforming L2P and DualPrompt. The method effectively balances plasticity and stability, highlighting language-informed prompting as a promising direction for robust continual learning

**Strengths:**

1. Paper uses language-guided prompt selection that’s trained once, used visually at test time. CoLaP aligns a visual selector to language embeddings via contrastive + distillation losses, then drops text at inference preserving efficiency while improving selection robustness.

2. It clusters auto-generated captions to form prompt keys, encouraging concept sharing across related classes and reducing interference.

3. OOD robustness and Consistent gains on ImageNet-O, Oxford-IIIT-Pet, etc., showing >5% improvements over L2P in key OOD settings.

4. Maintains stability as tasks increase (5→20) with sensible top-K prompt retrieval.

**Weaknesses:**

1. While the specific combination of contrastive alignment + discrete prompt keys is new, its conceptual ingredients language guidance, multimodal contrastive learning, and prompt tuning are well explored (e.g., LGCL ICCV 2023, Roy CVPR 2024, Progressive Prompt ICLR 2023)

2. Limited baselines: The paper claims SOTA but omits contemporary multimodal CL baselines (Roy 2024; LGCL 2023; PromptAlign NeurIPS 2024). Without these, the improvement claims can’t be trusted across modalities

3. The model highly dependent on the text caption, yet there’s no sensitivity or noise ablation. If captions are incorrect or generic, how robust is the contrastive alignment? This is a critical missing analysis since the approach’s strength hinges on textual fidelity.

4. The alignment network will get trained on the task specific data, training one after another task this network itself will suffer from forgetting how paper handle the same?

5. While discrete prompt indices reduce memory, they remove continuous similarity structure. This may harm fine-grained transfer or incremental compositional reasoning, provide the ablation on the same.

**Questions:**

Compare with the recent SOTA model, provide the answer discussed the point in the weakness section.

---

### Meta-Review · Area_Chair_jXvu · 2025-12-29

**Summary:**

All reviewers show a tendency to reject this paper due to the limited novelty and insufficient comparison of this paper. Moreover, the authors have not provided the rebuttal. The Meta reviewer carefully read this paper and the comments, which indeed contain many problems that remain unsolved, like the limited novelty, incomplete comparison, and overstated contributions. After considering the reviews, rebuttal, and the author's message, the Meta reviewer agrees with the concerns raised by the reviewers and recommends rejecting the paper.

**Reviewer Concerns:**

All concerns raised by the reviewers have not been addressed.

**Reviewer Scores:**

None

---

### Decision · Program_Chairs · 2026-01-26

Reject